# Perceived needs of patients and family caregivers regarding home-based enteral nutritional therapy in South Africa: A qualitative study

Nomaxabiso Mildred Mooi[ID]*[◷], Busisiwe Purity Ncama[ID][◷]

School of Nursing and Public Health, College of Health Sciences, University of KwaZulu-Natal, Durban, South Africa

◷ These authors contributed equally to this work.
* mooi@ukzn.ac.za

## Abstract

### Introduction

The need for specialized care, particularly enteral nutritional therapy in community settings is now increasing with implications for both patients and primary care providers. More research is needed to identify the needs of patients and primary caregivers. The study aimed to explore the perceived support needs regarding the provision of home-based enteral nutritional therapy among critically ill adult patients and family caregivers in the Kwa-Zulu-Natal Province of South Africa.

### Methods

A qualitative study of purposely selected adult patients on homebased enteral nutritional therapy and family caregivers was conducted in a district hospital, a community health centre, two primary health care clinics and selected households in the KwaZulu-Natal Province, South Africa. Semi-structured individual interviews were conducted between June and September 2018 and the content analysis approach was used to analyse data.

### Results

Two major themes and five subthemes emerged from the results of the interviews. The major themes concerned socioeconomic and psychosocial support needs related to the provision of home-based enteral nutritional therapy. Subthemes included the need for financial assistance, need for enteral nutrition products and supplementary supplies, need for infrastructure for continuity of care, and psychological support needs.

### Conclusion

Results of this study confirm the need for developing strategies adapted to a South African context and yonder to meet patients' and family caregivers' needs with regard to nutritional services. More research on the identification of needs through monitoring and evaluation of

**Data Availability Statement:** All relevant data are within the manuscript and its Supporting Information files.

**Funding:** The authors received no specific funding for this work.

**Competing interests:** The authors have declared that no competing interests exist.

the implementation of nutritional guidelines is needed, particularly in the district hospital and primary health care (PHC) setting.

## Introduction

People needing care post the intensive care unit (ICU) are now increasing in community settings, which has implications for both primary health care professionals and family caregivers [1]. Many patients who survive critical illness need a long time to recover fully due to vulnerability to malnutrition and the risk of infections [2]. Of concern is that illness-related malnutrition is recognised as the number one cause of death and disability warranting introduction of a good nutrition plan for nutritionally at risk patients discharged home or to primary health care institutions [3]. The increase in survivors of critical illness and the increased emphasis on the management of illness-related malnutrition in the community have resulted in an increased number of patients receiving home-based nutritional therapy [4–6]. However, optimising nutritional care post hospital discharge can be challenging. Evidence shows that, following discharge, there is insufficient information and support for patients and family caregivers to assist in nutritional therapy management, despite its widespread use [7, 8]. This is in spite of enteral nutritional therapy (EN) having positive effects on the clinical outcomes of critically ill patients and the capacity to have a transformative effect on patient and family life in the primary health care setting [9, 10].

Primary health care (PHC) reengineering and plans for implementing National Health Insurance (NHI) are strategies aimed at managing the global burden of both communicable and non-communicable diseases and resultant chronic illness [11, 12]. EN has been documented as a safe and cost-effective intervention for managing these diseases at all levels of care. Home-based enteral nutritional (HEN) in particular has distinctive clinical and social benefits, which may restore some independence to patients and their families [13]. However, management structures, funding challenges and the need for further education, particularly within the primary care setting, may limit optimal use of HEN [14]. In countries and health systems experiencing disproportionate prevalence and costs of non-communicable diseases (NCD) related malnutrition and associated prolonged hospital stay, the role of HEN is bound to expand [14]. Although multidisciplinary interventions and the development guidelines for successful discharge are designed to benefit all affected parties, unfulfilled support needs of patients and family caregivers of adults on HEN have been documented [15, 16]. The aim of this study, therefore, was to explore and describe perspectives on support needs of adult patients discharged from hospital on HEN and their families.

## Materials and methods

### Study design and setting

This study adopted a qualitative descriptive design, a design that allows for deeper understanding of phenomena under study [17, 18]. This study was useful in generating an in-depth understanding of the perspectives of patients on HEN and their family caregivers, individuals directly affected and specific to the context of South African health care. The study formed part of a mixed methods multiple case study of a district hospital and primary healthcare institutions in a district in KwaZulu-Natal Province, South Africa. The aim was to develop a model for implementation of guidelines for enteral nutritional therapy practice.

Harry Gwala District, one of the poorest districts in South Africa, is situated in the south of the KwaZulu-Natal Province and incorporates four public district hospitals and 39 clinics,

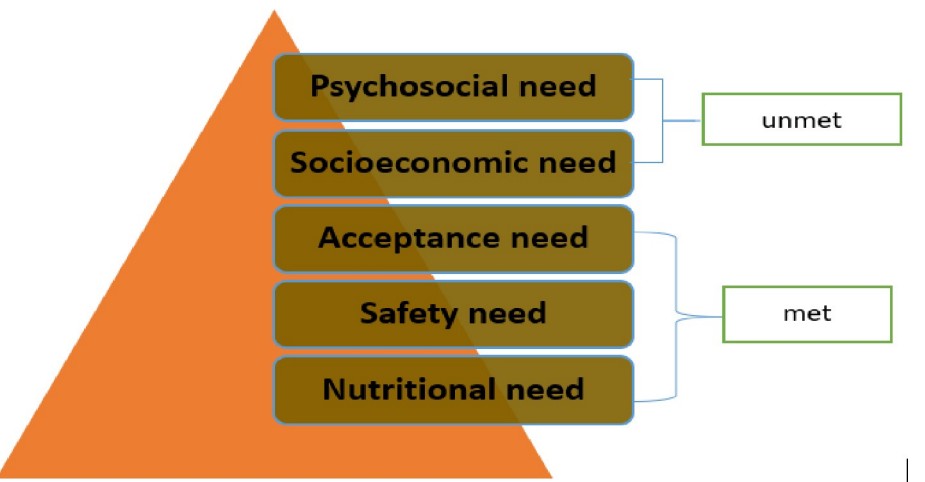

**Fig 1. Maslow's hierarchy adapted to home-based enteral nutritional therapy provision suggesting that, when no effort is made to meet lower needs, higher needs at the apex of the pyramid are seldom realised.**

with the latter mainly operated by nurses and a few doctors. The district has a population of 492 203, with a population density of 46.7 persons per km$^2$ falling in socio-economic Quintile 1 and thus counted among the poorest districts in South Africa [19]. The estimated medical scheme coverage is 5.9% [20]. The district does not have a tertiary hospital or any intensive care facility; patients in need of such care are referred to another district, however, having to be discharged back to their district and their families when still in need of EN or HEN. The researchers thus aimed to determine the understanding of health care support needs of adults in need or on HEN in a district hospital and primary care setting to inform the development of a model for implementation of the national enteral nutritional therapy practice guidelines.

## Theoretical framework of the study

The study was based on Maslow's Hierarchy of Needs model [21, 22]. The model was developed by Abraham Maslow [23] who identified five levels of need, each building on the other and arranged in a pyramid ranging from the lowest to the highest as: physiological needs, safety needs, belonging and love needs, esteem needs, and self-actualisation needs (Fig 1).

**Factors explanation.** The physiological needs basically include the need for food, air, and water, and fall on the lower level of the pyramid (level 1). The second level comprises security, stability, protection; freedom from fear, anxiety, and chaos, which are considered safety needs. Belonging and love needs occupy the third level of needs and involve the giving and receiving of love. The fourth level depicts societal recognition, the need for esteem, which is fulfilled by the ability to change one's environment, and respect. The fifth level presents the need for self-actualisation, which involves maximizing one's unique potential in life [24].

**Application to home based enteral nutritional therapy.** Each of Maslow's five levels of need has inferences for managing home-based enteral nutritional therapy for adults in a middle-income country. The hierarchy indicates the importance of fulfilling more basic needs first during critical illness. In our study we propose that diagnosing the need and initiating HEN on admission and on discharge from ICU and hospital is a first-order need. Consequently, any illness that can lead to malnutrition and critical illness can worsen such a state as patients may become immunocompromised and unable to avoid the inflammatory response, and therefore are prone to poor outcomes [25, 26]. The need for safety from EN complications including,

but not limited to, vomiting, aspiration, relocation, leakage and blockage of the feeding tube, and stoma site infections is the second order need [27]. The third level need in this study is likened to the feeling of being rejected due to decreased functionality and dependence on others for nutritional needs caused by chronic critical illness-related malnutrition, as well as the unnatural method of feeding. It is evident that special support systems, which can be mobilized by the multidisciplinary team, may be vital to address this need. At the fourth level, the inability of the patient and/or family caregiver to continue with daily activities and resulting unemployment can deteriorate socio-economic status. Lastly, the fifth level, self-actualization, can be achieved if the first four levels of the pyramid are fulfilled.

## Population, recruitment and sampling

The target population for this study comprised adult patients on HEN who have been discharged to community care and homes and family assigned caregivers. This was based on the assumption that they would be able to provide the required information which would inform the development of effective strategies to implement the national EN practice guidelines.

To be included in the study, a participant had to be 1) an adult, 18 years of age and older, 2) a patient on EN due to inability to swallow or feed themselves and needing the support of others to meet their nutritional needs, 3) a patient on HEN or percutaneous entero-gastrostomy (PEG) feeding, 4) a family caregiver or any other person assigned by the family for taking care of and staying with the patient. Paediatric patients on HEN and adult patients on EN in a tertiary hospital were excluded.

Potential participants were identified and recruited through tertiary hospitals where HEN was prescribed and initiated, and which were referral hospitals for the district under study (dietitian or stoma therapy nurse). Permission to access the dietetics and stoma therapy departments of the selected districts from which patients on HEN were discharged was sought from the respective district offices. The dietitians or stoma therapy nurses acted as recruiters in this study by providing the research information/documentation to prospective participants. Appointments for interviews of patients with capacity to give consent and family caregivers who showed interest were arranged by telephone. Non-probability purposive sampling based on data saturation was used to select the study sample.

## Data collection

Research data were collected between June and September 2018. Individual interviews were conducted using a semi-structured interview guide, which included six open-ended qualitative items based on the five levels of the Maslow's Hierarchy of Needs model (Fig 1). The items included: 1) description of the EN therapy (reason/indication and how it began), 2) education given on discharge (concerns of safety), 3) type of support needed, extent of limitations and related perceptions, 4) structural needs including financial resources, and 5) emotional support needs (psychosocial services need). A concluding question sought suggestions from interviewees on what could be done differently to meet their needs.

**Semi-structured interviews.** The researcher visited each household at least once before the interview to become familiar with people and surroundings and that helped to establish good rapport with prospective interviewees. Upon obtaining informed written consent, interviews during which only the researcher and the interviewer were present commenced in the selected households. Each interview took 30 to 45 minutes, although individual variations occurred. The researcher conducted the interviews personally to be able to take field notes while observing the setting and artefacts that could influence the interpretation of results. The interviews allowed for better exploration of individual opinions than would have been possible

with the use of group interviews. For instance, it was expected to be difficult for relatives to express themselves freely about how they felt about taking care of their relatives on HEN with them listening. The interviews were audiotaped and audio recordings were transcribed verbatim. Verbatim quotes were edited for clarity and ease of reading by using . . . for elimination of words or phrases.

## Data analysis

Data analysis was conducted concurrently with data collection using content analysis and following the steps proposed by Graneheim and Landman [28]. The recorded interviews were transcribed and exported into NVivo version 12 for storage and easy access. Transcriptions were read thoroughly and repeatedly, as recommended by Erlingsson and Brysiewicz [29]. Text from individual interviews was organised into meaning units that were condensed and coded. The codes were interpreted and compared for differences and similarities and sorted into subthemes in an attempt to answer the research questions. The second author confirmed the analysis. After a brief discussion the two authors agreed on five subthemes, and two major themes that unified the content in the themes finally were formulated. The descriptive themes were 'socioeconomic support needs' and 'psychosocial support needs'. These illuminated the needs of patients on HEN and family caregivers in the district hospital and PHC setting.

## Trustworthiness

Trustworthiness is an overarching concept encompassing several methods for describing aspects of trustworthiness in qualitative studies. These include credibility, dependability, confirmability, and transferability [30].

**Credibility.** To achieve credibility in this study, the researchers made an effort to find participants who had experienced the phenomenon under study and were able to tell about it, as recommended by Graneheim, Lindgren [30]. Participants were also asked to confirm information obtained during data collection interviews. Admission of researchers' beliefs, assumptions and recognition of limitations in the methods of the study and their potential effect are some of the important ways of increasing the credibility of data.

**Dependability.** The researcher (the first author) was open to discussion with the second author about their own pre-understanding, the way questions were asked, type of follow-up questions, and how the interviewees' narratives were perceived and interpreted. The second author suggested alternative interpretations that helped to address dependability. Sandelowski [31] recommends inclusion of more than one researcher in the analysis to address dependability as researchers' interpretative repertoires may differ.

**Confirmability.** The researcher's transparency concerning the interview process may have helped to increase confirmability of data and to ensure that the findings emerged from the data and not from the researcher's predisposition [32].

**Transferability.** Transferability is determined by the extent to which the results can be applied to other settings or populations [33]. To ensure transferability in this study, characteristics of the participants were clearly described and their responses were reported verbatim; this may present ideas to other researchers who conduct similar studies in future. Furthermore, research questions were based on existing theory and are attached as an appendix for potential use in similar studies.

## Ethical considerations

Ethics committee approval was gained from the University of KwaZulu-Natal Human and Social Sciences Research Ethics Committee and the KwaZulu-Natal Health Research and

Knowledge Management directorate with references HSS/1495/017D and HRKM413/17-KZ-201710-012 respectively. The two health district management offices gave permission for the researcher to access the participants. Written consent indicating participants' voluntary participation was obtained from the patients and family members who agreed to participate in the study after they had received information about the purpose of the study; how it would be conducted and that it would be recorded. They were further informed that they could withdraw from the study at any time. Pseudonyms were used throughout the study to maintain anonymity (P1-P7).

## Results

Sixteen people met the criteria for inclusion and showed interest in participating in the study, however, as some patients had their feeding tubes removed before the interview dates and some had died, this led to reduction in the number of participants. Eventually, three patients on HEN and four family caregivers were interviewed. All patient participants were males and three female relatives and one male participated. The age range of patients reported on was between 18 years and 76 years and the duration on HEN was between six months and 10 years, with one participant (P6) reporting the longest duration of 10 years. Data obtained from both patients and family caregivers showed that indications for HEN included cancer of the mouth and upper gastrointestinal tract, stroke and chemical injury of the upper gastrointestinal tract.

Participants verbalised that their needs on the first three of the five levels of Maslow's hierarchy of needs were fulfilled, citing their involvement in decision making about the feeding method, adequate education and training to prevent and manage possible complications. All participants reported good family and community support and acceptance, hence the focus of this study on the two major themes and five subthemes that emerged as unmet needs of patients and family caregivers regarding HEN provision.

### Theme 1: Socioeconomic support needs

Participants indicated that the provided financial support was insufficient to meet their domestic and social needs (PI, P2, P4, P5 and P7). P1 and P4 were concerned about the need for enteral nutrition feeding or formulas and food supplement supplies, which they were struggling to provide for their relatives because of the cost. They further stated that the formulas were suitable for diabetics and supplements were needed to achieve the desired weight for surgery for those who needed surgical interventions. P2 raised concern about the condition of his house, which he could not attend to because of financial constraints. The need for facilities other than tertiary institutions at which they could get assistance and resources for their feeding "infrastructure for continuity of care" was expressed by P1, P4 and P5. The summary of the subthemes under this theme is presented in Table 1.

### Theme 2: Psychosocial support needs

Four participants out of seven indicated the need for psychological support, mentioning that they were "psychologically drained", "frustrated", "worried", "hurt" and "feelings of giving up on self". Table 2 presents a summary of the theme 2 discussion together with their meanings and supporting quotations.

## Discussion

According to the results of the individual interviews with patients on HEN and their family caregivers, indications for HEN included cancer, stroke and domestic injuries. The findings

**Table 1. Summary of key subthemes under theme 1.**

| Subthemes | Representative quote |
|---|---|
| ***Need for financial assistance*** | *. . . . they said because my grandpa is already getting old age pension grant from government so they can't open a grant for sickness. (P5)* |
| | *. . .Increase the grant money, it's not enough, we are struggling . . .it is not enough . . .I need money for transport to hospital and I have to buy food for all of us. . .no, no, no, not enough at all. I also need money to fix this house . . . this house needs plastering, look, it is cold and humid . . . it is not good for my throat and chest, . . ., it makes me sick. (P2)* |
| | *I had to ask the doctor to go there once a month., transport money issue is killing me in the pocket, and his food is costly . . . is using a lot of milk to blend his food . . .. . . .1.8 kilograms only lasts for 2 weeks and is expensive . . . but what can I do! (Shaking head and shrugging shoulders). There. . .., there is this type of feed they say is good I should buy from XXXX store, I have not bought it, am still comparing prices, looking for a cheaper one . . . it is expensive and I don't have a proper job. (P1)* |
| | *. . .Diabetic formula G, 250mls is about R155, which sometimes her pension money does not cover (P4)* |
| | *I am not working full time, and I do not get pension money . . . (smiles) I am not that old. (P7)* |
| ***Need for enteral nutrition products and vitamin supplement supplies*** | *He eats from what we all eat, but cooks it his own way and blend it with the machine they gave him. . .his food cost a lot of money and is using a lot of milk to blend his food . . .. SSSS feed product, 1.8 only lasts for 2 weeks and is expensive. . . There is this type of feed they say is good I should buy. He also goes to another hospital for his vitamin injection . . . he needs vitamins to gain the weight they say he needs to be able to be operated on the throat. (P1)* |
| | *I go to this other hospital for injections. . . vitamin injections. (P2)* |
| | *Because she is diabetic, I was told to feed her diabetic formula D but I struggled to get it, she is now on diabetic formula G, which is also okay for diabetics. . .. (P4)* |
| ***Need for infrastructure for continuity of care*** | *His hospital is very far and . . . the clinic does not have the things he needs for the PEG, he also goes to the nearer hospital for his injections. He needs a school with a clinic inside. . . at the school here, they cannot take him. . ., scared that this tube can slip out. (P1)* |
| | *One time I took him to a private doctor . . . the hospital where this tube is treated is far, there they always say we must go where this tube was put in. (P5)* |
| | *I would expect to get home visits, I don't know. . . maybe twice a month, because she has stroke and it is even difficult to get her out of bed, to take her to hospital, because she can't even sit, if we could get somebody professional, maybe they can give us advise on how to handle this, this is draining us psychologically, maybe if we can get advice on how to cope with this. (P4).* |

are consistent with what Halliday, Baker [34] found, namely that HEN was common in patients diagnosed with upper gastrointestinal (GI) cancer who could not meet nutrition requirements with oral intake. The increased prevalence of cancer survivors furthermore is attributed to the development of more effective modern oncology therapies and advances in critical care [35]. Again, patients and/or family caregivers told us that they were informed or involved in decision making regarding the method of nutritional therapy. This was in line with what Kenny and Singh [36] argue; that decision making about enteral nutrition is often intricate and requires the consideration of a number of aspects, including respecting the wishes of the patient and their families, not only the medical need for the intervention. These authors further add that provision of artificial nutritional therapy can be an emotional topic, even for

**Table 2. Psychosocial support care needs regarding HEN.**

| Sub-themes | Frequency | Meaning | Evidence |
|---|---|---|---|
| Psychological support needs | 4 | Need for psychological services to intervene | . . . this is draining us psychologically, maybe if we can get advice on how to cope with this. (P4) |
| | | | I can see that he has not completely accepted his condition and way of eating, he still wants to eat normal food he cannot swallow . . . he gets frustrated . . . sometimes he refuses to eat. . ., as you can see, is losing weight, I am worried about him. (P3) |
| | | | Not many people have the kind of support I am getting from my family and are positive like me, some need a lot of psychological support, and it would be nice to get the care so that people don't just give up on themselves. (P7) |
| | | | this thing hurts me. . .. . .. maybe if their mother was still alive. . . you know my sister, it is hard. . .. (P1) |
| Social support needs | 2 | Inability to fulfil personal goals and need for social services to assist/relieve relatives looking after patients on PEG. | I don't know. . ., but if he can be able to continue with his education. . ., I am still waiting for social workers to find him a suitable school. . ., if it was not for this, he would have finished his matric by now. . ., when I go to work sometimes out of town, sometimes in the ZZZZ province, we have to ask his brother's girlfriend to assist him . . . .accompany him to hospital. . ., sometimes the neighbours. . .. (P1) |
| | | | Like. . .my life is on standstill. . ., I wanted to go and do a counselling course. . .. I have to be here for him (P3) |

health care professionals, which sometimes makes them uncomfortable and unsure of what recommendations to make [36]. The major themes that emerged from this study are socioeconomic and psychosocial needs.

## Socioeconomic support needs

In this study, financial support needs including transportation for follow-up care for which they expected support from KwaZulu-Natal health and social services were reported by five out of seven participants. Although the findings did not come as a surprise, as KwaZulu-Natal is described as one of South Africa's poorest and most densely populated provinces [37], it is a concern and measures have to be taken towards meeting the patients' needs. [36], Pillay and Skordis-Worrall [38] affirm the financial decline in public policy globally, meaning that South Africa is not an exception. This was supported by Holst and Rasmussen [3] who stated that the transition between hospital and home leaves patients and their relatives with unmet financial support needs. Two studies on rural cancer and HIV-positive patients also reported unmet financial needs [39, 40]. It is a fact that poor populations are the ones with the highest health risks and need for more health services, hence, the aim of financing for Universal Health Coverage (UHC) is that there are large pools of retained funds that can be used for their health care costs at times of need, regardless of their affordability [11].

Patient inability to pay for the transport costs of clinic visits has been identified as a leading cause of loss to follow-up and treatment default [41]. Kenny and Singh [36] and Wong, Goh, Banks, et al. [42] mentioned something different from other authors, pointing out that there are indirect costs that include taking time off work by family caregivers, as well as other intangible patient outcomes that need to be considered, besides the direct costs of HEN.

Participants also mentioned the need for supplies of special feeding formulas for certain conditions, for example, one family caregiver verbalized inability to afford to purchase a commercial formula that could last at least a month for a diabetic relative on HEN. This is one of the issues that need to be looked into, as recommendations of the national enteral nutritional therapy practice guidelines for adults include supplying special commercial formulas for patients with special conditions such as diabetes mellitus [43]. Regarding this, Ojo [27] posits

that the procurement and supply of HEN equipment and supplementary items necessitate an effective process to ensure uninterrupted delivery. This is essential because there should be continuity of service provision and enteral nutrition management when patients are discharged home from hospital. Weeks [44] adds that socioeconomic factors such as the financial situation, home sanitation, caregiver level of education, and motivation to follow preparation instructions closely need high consideration.

On the other hand, a positive finding was that patients reported being provided with food blending machines on discharge from hospital, which was good to note as in some countries there is a lack of adult training and support in managing HEN [14]. P1 and P2 also indicated that they needed to get vitamins and other blenderised tube feeding (BTF) supplements to achieve the target weights. Similar needs were echoed by Chen, Lai [45] who reported that a paediatric patient on HEN who was not put on vitamins was diagnosed with scurvy, a result of insufficient intake of vitamin C, which is easily preventable with nutritionally complete tube feeding. Living far away from the treating hospitals was mentioned several times in our study (P1, P2 and P6). In the United Kingdom, the delivery of feeding pumps to HEN patients was done by community nurses and the National Health Service (NHS) supplied them with the equipment or they received it directly from the manufacturer and costs were charged back to the hospital budget [27].

Regarding infrastructure and support care needs, some participants (P6 and P4) expressed the need for home visits by the dietician so that they could tell them about their feeding and PEG problems. In this study, this can be considered as a lack of HEN monitoring and is consistent with report from two studies that home visits and home-based nutritional services were restricted to certain patients [46] and most patients were relying on family caregivers for HEN provision [47]. The main concern was that the family caregivers often have limited relevant knowledge and training [46]. Unlike in other countries where a study by Boland, Maher [14], reported that the hospital dietitian was the primary source of support and monitoring, followed by a stoma therapy nurse and, to a lesser extent, by the community nutrition team.

There is a need for economic evaluation to calculate the cost of direct healthcare needed, including nutritional therapy provision, outpatient monitoring and management of complications and transportation processes related to nutritional therapy programmes [48]. Enteral nutritional therapy, particularly home-based EN, is a method of choice for artificial feeding, because it is more physiological, safer and cost-effective compared to the parenteral one [13]. It furthermore results in low incidence of morbidity [49]. However, recent reports have questioned the cost-effectiveness of HEN, which suggests that EN may be susceptible to overuse, particularly in long-term care settings.

### Psychosocial support needs

In this study, participants reported various psychosocial support needs of both patients and family caregivers. These results are consistent with other studies on caregivers of patients on HEN, which revealed that caregivers of such patients were at risk of experiencing feelings of being burdened [50] and that such feelings were associated with high levels of psychological distress and anxiety [51]. According to Fitch and Maamoun [52], these may include intense emotional distress, ineffective coping, and reduced quality of life and be attributable to unmet support care needs. Consequently, a number of approaches have been used to assess supportive care needs before they result in psychological support needs [52].

A previous study recommends interaction with patients to assess their specific needs to be able to individualise assistance accordingly [52]. Ahanotu, Ibikunle and Hammed [53] postulate that the functional dependence of patients on caregivers physically and emotionally

overloads family members, referring particularly to the mothers who frequently assume responsibility for the care provided to these patients, as their study was focused on children. Alananzeh, Levesque [54] elaborated to state that, in their study, these emotional feelings were expressed as anxiety, sadness, worry, depression, and fear, which was understandable in their case as participants in their study were not in their home country and lacked a sense of belonging.

According to Boland, Maher [14], psychological distress may sometimes be accompanied by reluctance among HEN patients to leave their homes. Reluctance to go out of the home environment may sometimes be combined with reluctance to express these psychological needs, which may sometimes be the reason for psychological and emotional burden not being identified or reported as low in some communities[36, 40]. Bjuresäter [55] associate increased psychological needs with lacking preparation before discharge and lacking support at home causing insecurity and uncertainty.

Again, in this study, one participant (P4) expressed the need for home visits, professional advice on how to handle the relative with stroke–they could not even get her out of bed and were psychologically drained. This seems to be closely related to what [36], Chen, Lai [45] reiterated, namely that patients with more severe symptoms and caregivers with less social support from family were more likely to have overall unmet supportive care needs. They recommended that these caregivers should be assessed on the discharge of their relative with regard to provision of psychological counselling and availability of a support group. Caregivers are the most involved in the care of patients. As such, they are seen as the second victims of the illness underlying the use of HEN because of the level of strain they are experiencing [53]. In most cases, they take this role under sudden and extreme circumstances, with minimal preparation and little guidance and support from healthcare systems, which should therefore be considered [53].

## Strengths and limitations of the study

To researchers' knowledge, this is the first study to inform understanding of the needs of adult patients and family caregivers on HEN in South Africa since the publication of national enteral nutritional therapy practice guidelines for adults. The study may inform the development of a model for implementation of the national guidelines and determine their feasibility and acceptability in the district hospital and primary care setting in the South African context. The individual interviews provided a platform for each care recipient to freely voice their personal opinions. The use of purposive sampling and the fact that the study was conducted in a single setting and with a smaller number of participants present the main limitations in this study. However, the characteristics of the actual sample was what was required to address the research questions and this matters more in qualitative research than the size of the sample, as suggested by Guetterman [56]. In addition, no inter-rate reliability was calculated following content analysis, however there was good agreement in coding and establishing representative themes between the researcher and second author.

## Conclusion

The study confirmed that adult patients on HEN and family caregivers within the district and PHC have socioeconomic and psychosocial needs that need attention if we are to succeed in providing nutritional care to all South African citizens. However, despite the unmet needs with regard to the last two levels of Maslow's pyramid, basic needs falling in the first three levels were fulfilled, which was a positive finding as the concept of home-based enteral nutritional therapy is still in its adolescent stage in South Africa. The study suggests nutritional therapy

practice integration through interdepartmental referral and collaboration with non-governmental and private partners to provide in structural and psychosocial needs of patients and family caregivers. There is also a need for education of primary health care professionals in the district where patients are returning home, perhaps provided by professionals at the tertiary sites. The study recommends continued support including home visits as indicated by participants. There is a need for more research on identification of needs through monitoring of patients and providing information regarding access to available resources. It would be of benefit to look at the tertiary institutions where such nutritional intervention is initiated, the discharge planning, and the referral system.

## Supporting information

**S1 Appendix. Interview guide for patient.**
(DOCX)

**S2 Appendix. Interview guide for family caregiver.**
(DOCX)

## Acknowledgments

The authors would like to acknowledge the healthcare professionals who assisted in the recruitment of participants and express gratitude to patients and family caregivers who participated in this research.

## Author Contributions

**Conceptualization:** Nomaxabiso Mildred Mooi, Busisiwe Purity Ncama.

**Data curation:** Nomaxabiso Mildred Mooi.

**Formal analysis:** Nomaxabiso Mildred Mooi.

**Investigation:** Nomaxabiso Mildred Mooi.

**Methodology:** Nomaxabiso Mildred Mooi.

**Project administration:** Nomaxabiso Mildred Mooi.

**Resources:** Nomaxabiso Mildred Mooi.

**Software:** Nomaxabiso Mildred Mooi.

**Supervision:** Busisiwe Purity Ncama.

**Validation:** Busisiwe Purity Ncama.

**Writing – original draft:** Nomaxabiso Mildred Mooi.

**Writing – review & editing:** Nomaxabiso Mildred Mooi, Busisiwe Purity Ncama.

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
