## [Decision Letter · Decision Letter 0]

9 Oct 2019

PONE-D-19-18633

Perceived needs of patients and family caregivers regarding home-based enteral nutritional therapy in South Africa: A qualitative study

PLOS ONE

Dear Dr Mooi,

Thank you for submitting your manuscript to PLOS ONE. After careful consideration, we feel that it has merit but does not fully meet PLOS ONE’s publication criteria as it currently stands. Therefore, we invite you to submit a revised version of the manuscript that addresses the points raised during the review process.

In addition to addressing reviewer comments, I recommend a review of the manuscript for language quality.  The language of a paper is difficult to understand in some sections.  In accordance with PLOS One guidelines, I recommend that the authors seek independent editorial help before submitting a revision. These services can be found on the web using search terms like “scientific editing service” or “manuscript editing service.”

We would appreciate receiving your revised manuscript by 9 November 2019. To enhance the reproducibility of your results, we recommend that if applicable you deposit your laboratory protocols in protocols.io, where a protocol can be assigned its own identifier (DOI) such that it can be cited independently in the future. For instructions see: http://journals.plos.org/plosone/s/submission-guidelines#loc-laboratory-protocols

We look forward to receiving your revised manuscript.

Kind regards,

Rosemary Frey

Academic Editor

PLOS ONE

**Journal Requirements:**

**Comments to the Author**

1. Is the manuscript technically sound, and do the data support the conclusions?

Reviewer #1: Partly

Reviewer #2: Yes

2. Has the statistical analysis been performed appropriately and rigorously? 

Reviewer #1: I Don't Know

Reviewer #2: N/A

3. Have the authors made all data underlying the findings in their manuscript fully available?

Reviewer #1: No

Reviewer #2: Yes

4. Is the manuscript presented in an intelligible fashion and written in standard English?

Reviewer #1: No

Reviewer #2: Yes

5. Review Comments to the Author

Reviewer #1: This is a good article, but reading extremely difficult, I had to read paragraphs multiple times to understand what you mean, I suggest given to a colleague with a fresh eye for a good rewrite . It need language editing, some sentences very long and some start with ''OF''. You are not allowed to mention product names or retailers in your article, it is against the Code. References are only at the end of sentences, do they say all exactly the same? If you defined abbreviation, used it in the rest of the article, do no write full words again.

Please read the following comments per line as in the PDF version I received:

2 = The definition for critical care = care of a patient in a life-threatening situation of an illness. Includes artificial life support system. = You need to remove the word critical illness / care from your article, the patients were not in ICU. You can say chronically ill / post ICU care . CHECK THE WHOLE ARTICLE FOR THIS

4 = rewrite = long to recover fully due to vulnerability to malnutrition and risk for infections.

20 = EN [already used before. do not write out again]

54 = Malsow mentioned, add Figure 1 after this paragraph, not only line 84

60 =add 'level 1'' in brackets

67 & 94 =HEN

89 = reword = doesn't make sense

92= [patient who had suffered one or more incidents of organ failure including the gastrointestinal tract and needed the support of others to meet their nutritional needs]... = GIT failure will mean no functioning GIT and TPN, so reword this. Patients are fed with PEG or HEN due to inability to feed themselves / cannot swallow etc = REWORD paragraph

98 = recruited = add in brackets [dietitian /stoma sister]

101-3 = CHANGE sentence = The dietician or stoma therapy nurse acted as recruiters in this study by providing the research information/documentation to prospective participants

106 - 111 = MUST move to RESULTS: [Sixteen potential participants met the inclusion criteria and showed interest in participating in the study; seven patients and nine family caregivers. Before the dates set for data collection, some patients had their feeding tubes removed and some had died, leading to appointments being cancelled. Eventually, three patients and four family caregivers selected through non-probability purposive sampling and based on data saturation were included in the study]

113-114 = REMOVE sentence [ After obtaining ethical clearance and approval for the study from the University and the provincial Department of Health]

114 = why only 3 months for data collection please explain.

125 [good rapport and to win the trust] = is this not the same thing? Reword

126 = verbal or written consent?

124, 12 = You mention ''One researcher'' and now ''The researcher'' did you use only one or more? Please check and change.

138 = DELETE [which was uploaded into the computer software program].

140 = ''celaining of data'' =please explain what you mean with this

142 = ço-coder'' = please define / explain

142-147 = REWRITE this, it is very confusing, I am still not sure how you analysed the data [A co-coder was engaged in coding the relevant information in the data and codes were sorted into categories in an attempt to answer the research questions. Analysis continued through creating themes to address the research question. Electronic files for each theme were then created and labelled, allowing for ease of access and management of data. Discussion and interpretation of the findings commenced thereafter]

Did you use any statistical software program or was it done manually?

164-5 = did you use one or more researcher? Please explain why if you did or didn't

188 = insert line 106-111 here

208 = PLEASE explain what you mean with this [ The need for infrastructure for continuity of care was expressed by P1, P4 and P5]

210-2 = remove commercial names of products and retailers [Glucerna, Diaben ,Nespray, Clicks]

259, 280 = I suggest you contact some other provinces Dep of Health, because in gauteng they do provide supplements at Clinics if patient qualify according to TTO guidelines and there are some home-visits groups.

270 = BTF = please define

340 = Please explain on what basis, you say your study was representative of study group. Can you honesty say 7 is enough? Also explain here why the study was not extended after the 3 months when you were only left with 7. This is for me a limitation

Reviewer #2: Very nice and well done qualitative study. Integration with Maslow's hierarchy is an excellent approach.

I have only a few questions and comments.

1) Was intercoder/interrater reliability (kappa score) calculated, since there was a primary and co-coder?

2) Do you have data regarding the total population of HEN patients in the district or community from which your purposive sample was identified? If so, please provide.

3) Theme 1: Socioeconomic support needs, line 202 -- do you mean "was not sufficient" or "was insufficient". Incorrect to say "was not insufficient".

For your information, you may find the following paper and resource helpful in your work.

Thompson CW, Durrant L, Barusch A, Olson L. Fostering coping skills and resilience in home enteral nutrition (HEN) consumer. Nutr Clin Pract 2006;21(6):557-565

http://www.copingwell.com/copingwell/HENCopingManual.pdf

6. PLOS authors have the option to publish the peer review history of their article (what does this mean?). If published, this will include your full peer review and any attached files.

Reviewer #1: No

Reviewer #2: No

---

## [Author Response · Author response to Decision Letter 0]

17 Nov 2019

RESPONSE TO REVIEWERS

Comments from Reviewers AUTHOR RESPONSE

Reviewer #2:

Very nice and well done qualitative study. Integration with Maslow's hierarchy is an excellent approach. I have only a few questions and comments. 

Thank you for the opportunity to revise my paper, your comments made a lot of sense contributed a great deal in shaping my paper.

1) Was intercoder/interrater reliability (kappa score) calculated, since there was a primary and co-coder?

 No, it was deemed unnecessary since the co-coders can bring varied but valid perspectives to identify unique codes and themes in data. Additionally, we were not for generalizability of findings beyond the sample to the entire population as we had a small sample size.

2) Do you have data regarding the total population of HEN patients in the district or community from which your purposive sample was identified? If so, please provide. No, we do not have data regarding the total population of HEN patients as it has been mentioned in the study setting section that the district under study was referring HEN candidates to tertiary institutions in other districts. It was therefore difficult to keep record.

3) Theme 1: Socioeconomic support needs, line 202 -- do you mean "was not sufficient" or "was insufficient". Incorrect to say "was not insufficient". We meant was insufficient, thank you for pointing that out. 

For your information, you may find the following paper and resource helpful in your work. Thompson CW, Durrant L, Barusch A, Olson L. Fostering coping skills and resilience in home enteral nutrition (HEN) consumer. Nutr Clin Pract 2006;21(6):557-565

http://www.copingwell.com/copingwell/HENCopingManual.pdf

Thank you for the information, it was helpful.

Reviewer #1: 

This is a good article, but reading extremely difficult, I had to read paragraphs multiple times to understand what you mean, I suggest given to a colleague with a fresh eye for a good rewrite

2 = The definition for critical care = care of a patient in a life-threatening situation of an illness. Includes artificial life support system. = You need to remove the word critical illness / care from your article, the patients were not in ICU. You can say chronically ill / post ICU care . CHECK THE WHOLE ARTICLE FOR THIS

4 = rewrite = long to recover fully due to vulnerability to malnutrition and risk for infections.

20 = EN [already used before. do not write out again]

54 = Malsow mentioned, add Figure 1 after this paragraph, not only line 84

60 =add 'level 1'' in brackets

67 & 94 =HEN

89 = reword = doesn't make sense

92= [patient who had suffered one or more incidents of organ failure including the gastrointestinal tract and needed the support of others to meet their nutritional needs]... = GIT failure will mean no functioning GIT and TPN, so reword this. Patients are fed with PEG or HEN due to inability to feed themselves / cannot swallow etc = REWORD paragraph

98 = recruited = add in brackets [dietitian /stoma sister]

101-3 = CHANGE sentence = The dietician or stoma therapy nurse acted as recruiters in this study by providing the research information/documentation to prospective participants

125 [good rapport and to win the trust] = is this not the same thing? Reword

126 = verbal or written consent?

138 = DELETE [which was uploaded into the computer software program].

142-147 = REWRITE this, it is very confusing, I am still not sure how you analysed the data [A co-coder was engaged in coding the relevant information in the data and codes were sorted into categories in an attempt to answer the research questions. Analysis continued through creating themes to address the research question. Electronic files for each theme were then created and labelled, allowing for ease of access and management of data. Discussion and interpretation of the findings commenced thereafter]. Did you use any statistical software program or was it done manually?

Comment noted and a language editor has been consulted. 

The definition for critical care = care of a patient in a life-threatening situation of an illness. Includes artificial life support system. = You need to remove the word critical illness / care from your article, the patients were not in ICU. You can say chronically ill / post ICU care . The whole article has been checked for this

Statement re-written as suggested.

Enteral nutrition written as EN.

Figure 1 added after the paragraph as recommended.

'level 1' added in brackets after pyramid.

HEN used in lines 67 and 94 which are 

Re-worded to “a patient fed with PEG or HEN due to inability to swallow or feed themselves and needed the support of others to meet their nutritional needs”.

“dietitian or stoma therapy nurse” added in brackets.

Sentence changed to “The dietician or stoma therapy nurse acted as recruiters in this study by providing the research information/documentation to prospective participants”

It is, if you have good rapport with someone, you two work with trust and sympathy, it was an error and the latter has been removed, thank you.

Written consent.

Deleted.

Statement has been rewritten to avoid confusion.

Please read the following comments per line as in the PDF version I received:

 Corrections were done per line as in the PDF we downloaded

It need language editing, some sentences very long and some start with ''OF''. Language editor consulted

You are not allowed to mention product names or retailers in your article, it is against the Code. That has been corrected and thank you for the comment.

References are only at the end of sentences, do they say all exactly the same? Efforts have been made to avoid putting all references at the end of statements as is if they were saying exactly the same thing; 273-276 and 291-294.

If you defined abbreviation, used it in the rest of the article, do no write full words again.

 Comment noted and the abbreviation issue has been corrected throughout the document.

89 = reword = doesn't make sense Reworded.

92= [patient who had suffered one or more incidents of organ failure including the gastrointestinal tract and needed the support of others to meet their nutritional needs]... = GIT failure will mean no functioning GIT and TPN, so reword this. Patients are fed with PEG or HEN due to inability to feed themselves / cannot swallow etc = REWORD paragraph

 Statement reworded and thank you.

98 = recruited = add in brackets [dietitian /stoma sister]

 Added

101-3 = CHANGE sentence = The dietician or stoma therapy nurse acted as recruiters in this study by providing the research information/documentation to prospective participants Sentence changed

106 - 111 = MUST move to RESULTS: [Sixteen potential participants Moved to results section

113-114 = REMOVE sentence [ After obtaining ethical clearance and approval for the study from the University and the provincial Department of Health]

 Removed

114 = why only 3 months for data collection please explain. It was due to time constraints on the researcher’s side. 

125 [good rapport and to win the trust] = is this not the same thing? Reword Reworded and thank you.

124, 12 = You mention ''One researcher'' and now ''The researcher'' did you use only one or more? Please check and change.

 There was one researcher, the first author and a project supervisor, the co-author, it checked and changed throughout the study.

138 = DELETE [which was uploaded into the computer software program]. Deleted

140 = ''celaining of data'' =please explain what you mean with this This was referring to manually and informally identifying and removing errors, incomplete words and statements from the transcripts before analysis. There were no professional tools or software used for that where we can trace the steps, as such, it has been removed if it mentioning it leaves the reader with high expectations. 

142 = ço-coder'' = please define / explain This was meant to be the co-author who was involved to confirm findings.

142-147 = REWRITE this, it is very confusing, I am still not sure how you analysed the data [A co-coder was engaged in coding the relevant information in the data and codes were sorted into categories in an attempt to answer the research questions. Analysis continued through creating themes to address the research question. Electronic files for each theme were then created and labelled, allowing for ease of access and management of data. The first author conducted the initial analysis and coding and the second author confirmed the findings. The section has been rewritten to make sense.

164-5 = did you use one or more researcher? Please explain why if you did or didn't

 One researcher, the first author collected data and second author (research supervisor) confirmed data analysis findings.

188 = insert line 106-111 here Line 106-111 inserted in the results section.

208 = PLEASE explain what you mean with this [ The need for infrastructure for continuity of care was expressed by P1, P4 and P5]

 The needed facilities for follow-up care either than tertiary hospitals where their PEGs were inserted.

210-2 = remove commercial names of products and retailers [Glucerna, Diaben ,Nespray, Clicks]

 [Glucerna, Diaben ,Nespray, Clicks replaced with diabetic formula, XXXX feed product and YYYY store.

259, 280 = I suggest you contact some other provinces Dep of Health, because in gauteng they do provide supplements at Clinics if patient qualify according to TTO guidelines and there are some home-visits groups. Comment noted and appreciated, but we feel it is a bit late for our study at this stage. However, it would be useful if we could include in our discussion section that other provinces in South Africa do provide the BTF supplements in the PHC clinics, unfortunately we could not locate any published material on that.

270 = BTF = please define

 BTF stated in full as Blenderised tube feeding, BTF in brackets.

340 = Please explain on what basis, you say your study was representative of study group. Can you honesty say 7 is enough? Also explain here why the study was not extended after the 3 months when you were only left with 7. This is for me a limitation Representativeness was referring to characteristics of the individuals that were required to address the research question than adequacy of the sample size. However, to avoid giving our readers an impression that we meant that 7 was enough to represent the entire population, the statement has been rephrased and stated as more of a limitation.

---

## [Decision Letter · Decision Letter 1]

12 Dec 2019

PONE-D-19-18633R1

Perceived needs of patients and family caregivers regarding home-based enteral nutritional therapy in South Africa: A qualitative study

PLOS ONE

Dear Dr Mooi,

Thank you for submitting your manuscript to PLOS ONE. After careful consideration, we feel that it has merit but does not fully meet PLOS ONE’s publication criteria as it currently stands. Therefore, we invite you to submit a revised version of the manuscript that addresses the points raised during the review process.

Please respond to the minor issues raised by Reviewer 2.

We would appreciate receiving your revised manuscript by 12 Jan 2020. To enhance the reproducibility of your results, we recommend that if applicable you deposit your laboratory protocols in protocols.io, where a protocol can be assigned its own identifier (DOI) such that it can be cited independently in the future. For instructions see: http://journals.plos.org/plosone/s/submission-guidelines#loc-laboratory-protocols

We look forward to receiving your revised manuscript.

Kind regards,

Rosemary Frey

Academic Editor

PLOS ONE

Reviewers' comments:

Reviewer's Responses to Questions

**Comments to the Author**

1. If the authors have adequately addressed your comments raised in a previous round of review and you feel that this manuscript is now acceptable for publication, you may indicate that here to bypass the “Comments to the Author” section, enter your conflict of interest statement in the “Confidential to Editor” section, and submit your "Accept" recommendation.

Reviewer #1: All comments have been addressed

Reviewer #2: (No Response)

2. Is the manuscript technically sound, and do the data support the conclusions?

Reviewer #1: Yes

Reviewer #2: Yes

3. Has the statistical analysis been performed appropriately and rigorously? 

Reviewer #1: Yes

Reviewer #2: N/A

4. Have the authors made all data underlying the findings in their manuscript fully available?

Reviewer #1: Yes

Reviewer #2: Yes

5. Is the manuscript presented in an intelligible fashion and written in standard English?

Reviewer #1: Yes

Reviewer #2: Yes

6. Review Comments to the Author

Reviewer #1: You did good work in addressing all the questions and shaping this paper! It read easy and you now understand fully what your goal was and your results. I think this is now a very good paper and you can be proud! Well done.

Reviewer #2: Most of the comments have been satisfactorily addressed. Even though you had a small sample, it is acceptable for a qualitative study in which the aim is to explore and describe perspectives ... and gain deeper understanding of phenomena under study; however, qualitative papers are enhanced by establishing inter-rater reliability, whether or not generalizability is limited. Please add a statement in the limitation section (lines 324) that no inter-rater reliability was calculated however there was good agreement in coding and establishing representative themes between the researcher and second author. I understand the difficulty in establishing population data given the lack of infrastructure in the district, but it is difficult for the reader to appreciate the importance of need for HEN patients without some sense of incidence/prevalence of condition. Could you explore whether data are available for # of patients discharged with HEN in the KwaZulu-Nata Province or any of the tertiary referral hospitals? IF none, please add a statement in the limitation section that there are no data on incidence/prevalence of HEN. In the conclusion, please address the need for education of primary care professionals in the district where patients are returning home, perhaps provided by professionals at the tertiary sites. The following sentences were unclear and may need rewriting: Line 7 - change nutritional risk to nutritionally at risk. Lines 178-180, lines 197-198, lines 273-276. Consider editing the verbatim quotes by eliminating expressions (yhoo!) (man) etc and inserting clarifying words when missing. You can add a statement in the methods section that says something like verbatim quotes were edited for clarity and ease of reading by using ... for elimination of words/phrases or insertion of [ ] to represent added words. Would also clarify what is PEGI (page 10).

7. PLOS authors have the option to publish the peer review history of their article (what does this mean?). If published, this will include your full peer review and any attached files.

Reviewer #1: No

Reviewer #2: No

---

## [Author Response · Author response to Decision Letter 1]

11 Jan 2020

RESPONSE TO REVIEWERS

Comments from Reviewers AUTHOR RESPONSE

Reviewer #1: You did good work in addressing all the questions and shaping this paper! It read easy and you now understand fully what your goal was and your results. I think this is now a very good paper and you can be proud! Well done. 

 I thank the reviewers for their constructive feedback and comments, they added value to my work.

Reviewer #2: Most of the comments have been satisfactorily addressed. Even though you had a small sample, it is acceptable for a qualitative study in which the aim is to explore and describe perspectives ... and gain deeper understanding of phenomena under study; however, qualitative papers are enhanced by establishing inter-rater reliability, whether or not generalizability is limited. 

 Thank you for the comment and it is noted for future purposes.

Please add a statement in the limitation section (lines 324) that no inter-rater reliability was calculated however there was good agreement in coding and establishing representative themes between the researcher and second author. I understand the difficulty in establishing population data given the lack of infrastructure in the district, but it is difficult for the reader to appreciate the importance of need for HEN patients without some sense of incidence/prevalence of condition. 

 Statement added and thank you.

Could you explore whether data are available for # of patients discharged with HEN in the KwaZulu-Nata Province or any of the tertiary referral hospitals? IF none, please add a statement in the limitation section that there are no data on incidence/prevalence of HEN. 

 There are no data on incidence/prevalence of HEN in adults in the province or any of the tertiary institutions and that has been added in the limitation section . 

In the conclusion, please address the need for education of primary care professionals in the district where patients are returning home, perhaps provided by professionals at the tertiary sites. 

 Thank you for the comment, a statement is added to the conclusion section.

The following sentences were unclear and may need rewriting: Line 7 - change nutritional risk to nutritionally at risk. 

Lines 178-180, lines 197-198, lines 273-276. 

 Line 7 changed to nutritionally at risk.

Lines 178-180, 197-198 and 273-276 rewritten for clarity. 

Consider editing the verbatim quotes by eliminating expressions (yhoo!) (man) etc and inserting clarifying words when missing. You can add a statement in the methods section that says something like verbatim quotes were edited for clarity and ease of reading by using ... for elimination of words/phrases or insertion of [ ] to represent added words. 

 Expressions eliminated and a statement has been added in the methodology section.

Would also clarify what is PEGI (page 10). Corrected to PEG, which has been written in full in its initial mention.

---

## [Decision Letter · Decision Letter 2]

28 Jan 2020

Perceived needs of patients and family caregivers regarding home-based enteral nutritional therapy in South Africa: A qualitative study

PONE-D-19-18633R2

Dear Dr. Mooi,

We are pleased to inform you that your manuscript has been judged scientifically suitable for publication and will be formally accepted for publication once it complies with all outstanding technical requirements.

With kind regards,

Rosemary Frey

Academic Editor

PLOS ONE

Additional Editor Comments (optional):

Please edit lines 335-339 as requested by reviewer.

Reviewers' comments:

Reviewer's Responses to Questions

**Comments to the Author**

1. If the authors have adequately addressed your comments raised in a previous round of review and you feel that this manuscript is now acceptable for publication, you may indicate that here to bypass the “Comments to the Author” section, enter your conflict of interest statement in the “Confidential to Editor” section, and submit your "Accept" recommendation.

Reviewer #2: All comments have been addressed

2. Is the manuscript technically sound, and do the data support the conclusions?

Reviewer #2: Yes

3. Has the statistical analysis been performed appropriately and rigorously? 

Reviewer #2: N/A

4. Have the authors made all data underlying the findings in their manuscript fully available?

Reviewer #2: Yes

5. Is the manuscript presented in an intelligible fashion and written in standard English?

Reviewer #2: Yes

6. Review Comments to the Author

Reviewer #2: Excellent revision. Please edit lines 335-336 and 338-339 on page 16 under strengths and limitations of the study as follows:

In addition, no inter-rate reliability was calculated following content analysis, however there was good agreement in coding and establishing representative themes between the research and second author.

Furthermore, there are no data on incidence or prevalence of HEN due to lack of infrastructure in the district, which can make it difficult for the reader to appreciate the importance for HEN patients.

7. PLOS authors have the option to publish the peer review history of their article (what does this mean?). If published, this will include your full peer review and any attached files.

Reviewer #2: No

---

## [Editor Report · Acceptance letter]

31 Jan 2020

PONE-D-19-18633R2 

Perceived needs of patients and family caregivers regarding home-based enteral nutritional therapy in South Africa: A qualitative study 

Dear Dr. Mooi:

I am pleased to inform you that your manuscript has been deemed suitable for publication in PLOS ONE. Congratulations! Your manuscript is now with our production department. 

With kind regards,

on behalf of

Dr. Rosemary Frey 

Academic Editor

PLOS ONE